

# Characteristics of muscle contraction of the rectus femoris using tensiomyography by sex in healthy college students: a cross-sectional study

Yasuaki Kusumoto[1], Hayato Goto[2], Kohei Chiba[3], Sakiko Oonishi[4] and Junko Tsuchiya[4]

[1] Department of Physical Therapy, Fukushima Medical University School of Health Sciences, Fukushima city, Fukushima, Japan
[2] LITALICO lnc, Setagaya-ku, Tokyo, Japan
[3] Department of Rehabilitation, Japanese Red Cross Medical Center, Shibuya-ku, Tokyo, Japan
[4] Major of Physical Therapy, Department of Rehabilitation, Tokyo University of Technology, Ohta-ku, Tokyo, Japan

Corresponding author
Yasuaki Kusumoto, kusumoto@fmu.ac.jp

## ABSTRACT

**Background:** Tensiomyography (TMG) is a non-invasive instrument for measuring mechanical muscle contraction characteristics and measuring the maximum displacement of the muscle belly in the radial direction with respect to the muscle and the time needed to achieve this from electrical stimulation. There have been only been a reports of TMG in healthy adults. A systematic review of TMG reported a low proportion of female participants, with a small sample size. Therefore, it is unclear whether there is a difference in TMG parameters according to sex and between dominant and non-dominant feet. Furthermore, the relationship between TMG parameters and evaluations commonly used in clinical practice has not been clarified. This study aimed to clarify the characteristics of muscle contraction of the rectus femoris using TMG according to sex among healthy college students and its relationship with muscle function evaluation, such as lower limb muscle mass and muscle strength.

**Methods:** This cross-sectional study included 91 healthy university students (18–24 years). Five tools were used: TMG, lower-limb muscle mass, rectus femoris thickness, isometric knee joint extension torque, and thigh circumference. Each parameter was compared by the generalized linear mixed model (GLMM) and Bonferroni's multiple comparison test, with sex as the without-subject factor and dominant/non-dominant foot as the within-subject factor. The correlation between the TMG parameters and other parameters was examined using Pearson's correlation coefficient for both males and females.

**Results:** The results of the GLMM, in terms of the TMG parameters, an interaction was observed for maximum displacement (Dm); in the results of the multiple comparison test, Dm for the non-dominant leg was significantly lower in females than in males. A main effect and interaction were not observed for delay time (Td) and contraction time (Tc) by sex, dominant foot, or non-dominant foot. There was a main effect of sex on muscle function evaluation parameters ($\rho \leq 0.05$). The correlation between TMG parameters for males and females and lower limb muscle mass, muscle thickness, joint torque, and thigh circumference were

![PeerJ]

significantly correlated with some TMG parameters, lower limb muscle mass and muscle thickness ($\rho \leq 0.05$). The absolute value of the correlation coefficient was low overall (0.20–0.38).

**Conclusion:** In healthy college students, TMG parameters for the rectus femoris showed sex differences in Dm, and there was a weak correlation between TMG parameters and lower limb muscle mass. TMG parameter evaluation may indicate a different function compared to the traditional muscle function assessment used in clinical practice. When using the Dm of the TMG as an evaluation battery for the rectus femoris muscle, it is important to consider sex-related differences.

## INTRODUCTION

Tensiomyography (TMG; TMG-100: TMG manufactured by TMG-BMC) is a non-invasive instrument for measuring mechanical muscle contraction characteristics. TMG determines muscle contraction time, muscle fatigue, and muscle fitness by measuring the maximum displacement of the muscle belly in the vertical direction (radial direction with respect to the muscle) and the time needed to achieve this from electrical stimulation (*Macgregor et al., 2018*). In recent years, TMG has been mainly utilized mainly in the field of sports in Europe. By understanding the characteristics of muscle contraction before and after training. It is also used to gauge the characteristics of muscle contraction among athletes in specific sports, and the effectiveness of training during specific periods or competition seasons to prevent, and hasten recovery from, injuries (*Loturco et al., 2016*; *Peterson & Quiggle, 2017*). According to a recent review on TMG, soccer and volleyball athletes have been the most studied (*García-García et al., 2019*). It has been concluded that TMG is useful for determining changes in muscle contraction characteristics between competition and training (*García-García et al., 2019*). A previous study of healthy adults between the ages of 35 and 90 found that while the correlation between Contraction Time and age was more significant for men than for women, multiple regression analysis with Contraction Time as the dependent variable did not extract sex, but age and exercise type (*Šimunič et al., 2018*). In short, regular endurance sports activity exacerbates age-related skeletal muscle slowing, and age-related slowing is found in all muscles regardless of discipline; however, information on sex differences in TMG parameters is lacking.

A previous longitudinal study with TMG in 9- to 14-year-old children reported no difference in the contraction time of the erector spinae between boys and girls; however, boys had a shorter biceps femoris contraction time than girls (*Simunic et al., 2017*). Additionally, the biceps brachii and vastus lateralis muscles have a higher contraction time in boys than in girls (*Simunic et al., 2017*). Biopsy data have confirmed that the biceps brachii in children have a higher percentage of type I fibers than adults (*Johnson et al., 1973*; *Pisot et al., 2004*) and are known to have longer contraction times–reflecting slower muscles—than adults (*Volmut, Pisot & Simunic, 2016*). Non-invasive TMG provides useful
information to validate sex differences in skeletal muscle contractile properties when direct measurement of skeletal muscle fiber-type composition is difficult for ethical reasons (*Simunič et al., 2011*). TMG can also be used for the early detection of muscle dysfunction due to bed rest before atrophy, and the architectural changes associated with obvious atrophy are detected by ultrasound (*Šimunič et al., 2019*). However, previous studies have differed in the presence or absence of sex differences between muscles, and exercise frequency has not been considered when analyzing TMG parameters.

Clinical assessments of muscle mass, thickness, strength, and circumference have been performed in all disciplines, and are the most common muscle function evaluations performed in many studies and general hospitals to evaluate the individual condition of the lower limb muscles (*Paes, Marins & Andreazzi, 2015*; *Lacio et al., 2021*). Sex-based differences in muscle mass, thickness, strength, and circumference were observed, with females having lower values than males. In healthy university students, the relationship between each parameter differs between men and women, as thigh circumference is positively correlated (0.54, 0.45) with muscle strength and muscle thickness in men and thigh circumference is positively correlated (0.54) with muscle thickness in women (*Kobayashi et al., 2018*). It is also generally known that these values do not vary significantly between the dominant and non-dominant feet within the same subject. However, a systematic review of TMG reported a low proportion of female participants with a small sample size (*Lohr et al., 2019*). Therefore, it is unclear whether there is a difference in TMG parameters according to sex and between dominant and non-dominant feet. Once the relationship between TMG parameters and evaluations commonly used in clinical practice is clarified, changes in muscle contraction characteristics in response to disease and treatment will become apparent, and the clinical application of TMG in many patients will advance. Therefore, it is necessary to investigate its relationship with various muscle function assessments and accumulate data on healthy individuals for it to be utilized among people with illnesses at general hospitals in the region.

Previous studies have reported TMG in various muscles (*García-García et al., 2019*). Among the various muscles, the neuromuscular function of the rectus femoris has been measured in many competitive sports and studies by training type (*Lohr et al., 2019*; *de Paula Simola et al., 2015*). Clinically, the rectus femoris muscle is one of the most frequently injured muscles in sports such as track and field, and soccer, in which repeated sprinting and kicking are used, both in professionals and amateurs, and has therefore received attention for some time (*McAleer et al., 2022*; *Armstrong, Pass & O'Connor, 2022*; *Mendiguchia et al., 2013*).

Therefore, this study aimed to clarify the muscle contraction characteristics of the rectus femoris using TMG in healthy college students by sex, and their relationship with common tests for muscle function used in clinical practice. We formulated two research hypotheses for this study. First, the maximum displacement of the TMG parameter differs by sex and between the dominant and non-dominant feet. In addition, the TMG parameters, delay time and contraction time, did not differ by sex and between the dominant and non-dominant feet. Second, the TMG parameters were significantly correlated with evaluations commonly used in clinical practice. In general, people with more exercise

experience tend to have a greater muscle mass and better neuromuscular function and performance. In Japan, many students engage in club activities up to high school, but few university students exercise on a continuous basis unless they are involved in club activities. Therefore, we hypothesized that students who do not exercise on a daily basis, even those with high muscle mass, would have stiff muscles and reduced contractility. Specifically, we hypothesized that lower limb muscle mass and muscle thickness would correlate negatively with the TMG parameter maximum displacement and positively with the TMG parameters delay time and contraction time. Knee joint extension torque correlates positively with the TMG parameter of maximum displacement and negatively with delay time and contraction time. Thigh circumference was uncorrelated with all TMG parameters.

## MATERIALS AND METHODS

### Participants

All the participants were enrolled at a university in Tokyo. The survey period was from April to May 2016. The inclusion criterion of participants was healthy university students with no underlying endogenous diseases. To determine the required sample size, a power calculation was conducted using G power, and the effect size was calculated using a moderate effect size based on Cohen's criterion (Cohen, 1988). In the generalized linear mixed model (GLMM), referring to the criterion in the two-way ANOVA, with the effect size set to 0.25, alpha level set to 0.05, and power set to 0.80, it was calculated that a total sample size of $n = 36$ was required. In the correlation analysis, with the effect size set to 0.30, alpha level set to 0.05, and power set to 0.80, a sample size of $n = 84$ was calculated. The participants were 105 healthy university students (aged 18–24 years) who provided written consent. The additional subjects were included to allow for missing values. Participants who underwent major trauma or surgery within 6 months, had a history of ligament injury around the knee joint, or were unable to complete the measurement were excluded. Accordingly, 91 participants who met the inclusion criteria were analyzed (Fig. 1). Dominant foot was defined as the foot used to kick the ball. The dominant foot was the right foot in 83 patients and the left in 8. The university where the study was conducted granted ethical approval to conduct the study within its facilities (authorization number: E16HS-030).

### Design

This was a cross-sectional study involving college students in Tokyo.

### Measures

The participants were instructed to perform no high-intensity exercise for 36 h before the measurements. The measurement parameters included TMG, lower-limb muscle mass determined using a body composition analyzer, rectus femoris thickness determined with an ultrasonic measuring device, isometric knee extension torque with a manual muscular strength meter as the maximum muscle strength, and thigh circumference. Height and weight were noted prior to the measurements. Considering the effect upon measurement

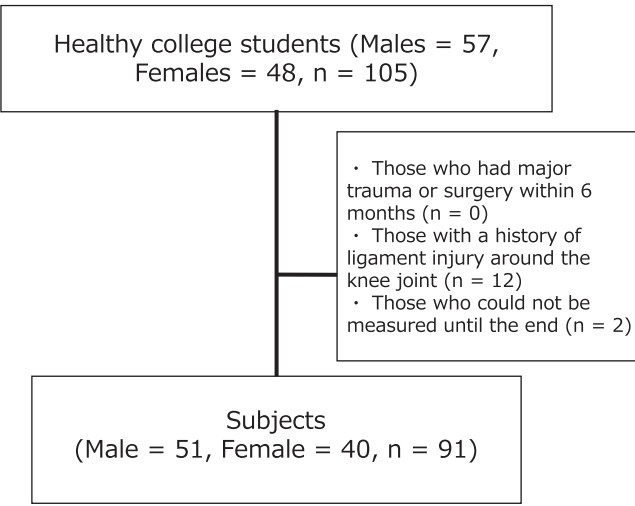

**Figure 1  Flow chart of the subjects.**   

parameters due to the measurement sequence, and the effects of fatigue due to long measurement time, a 2- to 3-min break was taken between each measurement. The order of each measurement parameter was randomly determined for each subject using a random number table created in Excel.

### TMG

The same investigator performed the TMG (TMG Measurement System; TMG-BMC Ltd., Ljubljana, Slovenia) on the rectus femoris in the dominant and non-dominant feet. All measurements were performed during electrically induced isometric contractions, in a resting supine position with the knee at an angle of 30° flexion (0° represents a fully extended knee joint), and an attached foam padding was used to support the knee joint in free. The TMG sensor rod was placed in the middle of the anterior inferior iliac spine to the upper edge of the patella, perpendicular to the belly of the rectus femoris (Fig. 2). Maximum displacement of the muscle belly in the radial direction with respect to the muscle was measured using a digital transducer, Dc–Dc Trans-Tek® (GK 40; Panoptik d.o.o., Ljubliana, Slovenia). Electrodes (TMG electrodes; TMG-BMC d.o.o. Ljubljana, Slovenia) for electrical stimulation (5 cm × 5 cm) were attached proximal and distal to the sensor rod, creating a 5-cm distance between the electrodes (*Tous-Fajardo et al., 2010*). The starting current was 20 mA, and the pulse stimulation time was 1 ms. The current was increased by 10 mA, and electrical stimulation was repeated until no changes in the waveform were observed or a maximum output of 110 mA was reached (*Rey et al., 2012*). To minimize the effects of enhancement and fatigue of the muscle, we included an inactive time of 10 s between each mA increment. The evaluation parameters were maximum displacement (Dm), delay time (Td), and contraction time (Tc). These have been reported to be highly reliable and recommended for use in previous studies (*Martín-Rodríguez et al., 2017*).

A typical TMG waveform is shown in Fig. 3. Dm, the vertical displacement of the radial waveform, expresses the amount of vertical displacement of the contracted muscle in

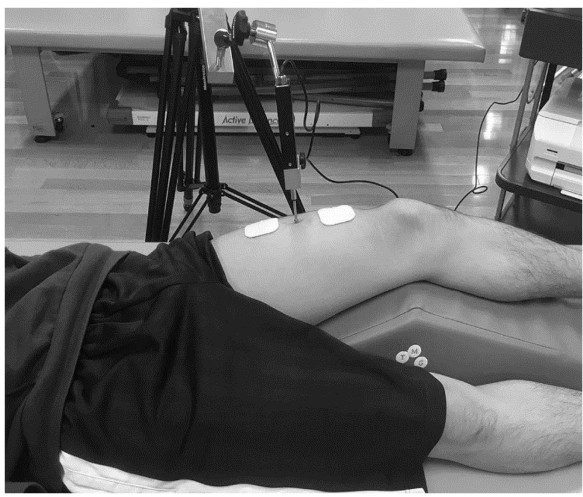

**Figure 2 TMG measurement in a participant.**

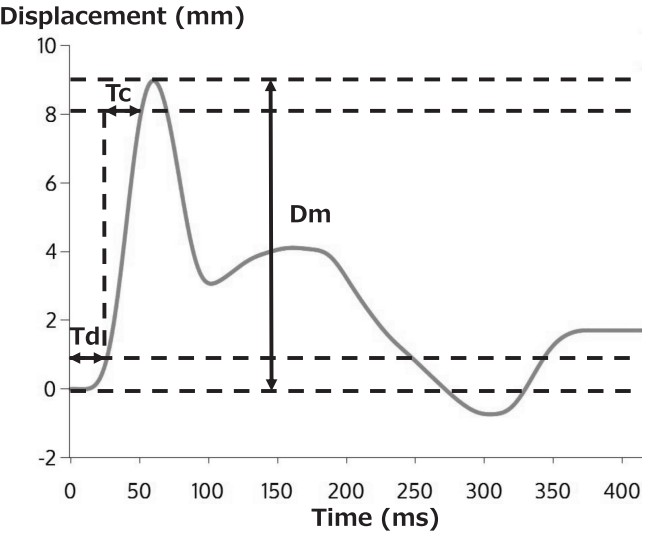

**Figure 3 Typical TMG measurement results.**

millimeters in response to electrical stimulation. A decrease in Dm corresponds to an increase in muscle stiffness (*García-García et al., 2019*; *Šimunič et al., 2019*). Td represents the time required (ms) to reach 10% of Dm after electrical stimulation, reflecting the responsiveness of muscles to muscle contraction due to electrical stimulation (*García-García et al., 2019*). Tc represents the time needed (ms) to achieve 10% to 90% of Dm, reflecting the contraction rate from the onset to the end of the muscle contraction. Therefore, Tc is said to be affected by muscle fiber type, and it increases as tendon stiffness decreases (*Simunič et al., 2011*; *Dahmane et al., 2005*).

To confirm the accuracy of the inspector's measurement technique, TMG was performed twice within 1 week in eight healthy university students who were not the subjects of this study. The intraclass correlation coefficients (ICC) (3,1) were calculated to be 0.75 for Dm, 0.61 for Td, and 0.74 for Tc.

### Body composition measurement

InBody S20 (InBody, Gangnam-gu, South Korea) was used to measure the body composition. With the participants resting in a supine position, a total of eight contact electrodes were attached, four points on the bilateral 1st and 3rd fingers, four points around the medial and lateral malleolus bilaterally, and five different frequencies (1, 5, 50, 250, 500 kHz). The skin of the measurement part was first cleaned with alcohol to remove the keratin on the skin surface. Of the left and right upper and lower limbs and trunk parameters measured by body composition measurement, skeletal muscle mass index was calculated from the muscle mass of the upper and lower limbs. In this study, the lower-limb muscle masses of the dominant and non-dominant legs were analyzed.

### Rectus femoris thickness

The rectus femoris thickness was measured in a resting supine position with the knee extended using an ultrasonic image-measuring device (ACUSON P300 and 18–6 MHz linear probe; Siemens Japan Co., Ltd., Shinagawa-ku, Japan). The measurement site was determined using a measuring tape and was marked with a permanent marker.
The measurement method was performed in accordance with previous studies, in which the probe was applied perpendicularly to the rectus femoris at the midpoint of the anterior superior iliac spine to the upper edge of the patella, and its angle was adjusted to maximize the view of the femur (*Tillquist et al., 2014*; *Kobayashi et al., 2018*). In addition, all measurements were performed by one investigator after sufficient practice, taking care not to compress the muscles to obtain good quality images. The image at this time was recorded on a computer, and the maximum distance of the transverse images of the rectus femoris muscle was measured in mm using the attached software and recorded as the thickness of the rectus femoris muscle. The rectus femoris thickness has sound reliability, and the ICC of measurements were 0.93–0.99 (*Kobayashi et al., 2018*).

### Knee joint extension torque

Knee joint extension torque was measured using a handheld dynamometer (Mutas F-1, ANIMA) by fixing the belt to the support of the bed while the subjects were seated with the hip and knee joints flexed at 90 (*van Trijffel et al., 2010*). The belt length was adjusted so that the lower leg was in the drooping position, and care was taken not to shift the sensor pad. The moment arm (m) was measured from the knee joint space to the center of the sensor. The subjects were instructed to maintain their trunk in a vertical position with both upper limbs folded in front. After warm-up with three-five explosive knee joint extension exercises, knee extension muscle strength was measured with maximum explosive effort for approximately 3 s. Breaks lasting 30 s or more were provided for the left and right legs, and each was performed twice. The maximum value was used, and the torque-to-body weight ratio (Nm/kg) was calculated. The knee joint extension torque has sound reliability, and the ICC of the measurements was 0.94 (*Hirano et al., 2020*).

### Maximum thigh circumference

The measurement was performed in the resting supine position with a knee joint extension of 0° using a cloth measure. The measurement method was performed in accordance with

**Table 1 Subject attributes.**

|  | Male (*n* = 51) | Female (*n* = 40) | *p* value | Effect size |
|---|---|---|---|---|
| Age, year | 21.2 (1.0) | 20.9 (1.2) | 0.15 | — |
| Height, cm | 172.3 (5.9) | 161.4 (16.4) | $p < 0.001^*$ | 11.69 |
| Weight, kg | 65.5 (8.8) | 50.3 (4.9) | $p < 0.001^*$ | 7.34 |
| SMI, kg/m$^2$ | 8.1 (0.8) | 6.0 (0.8) | $p < 0.001^*$ | 0.79 |
| Dominant foot (right, left), *n* | 47, 4 | 36, 4 | 0.72 | — |
| Exercise frequency of at least 1 h per day per week (0 days, 1–2 days, 3–4 days, 5 days or more; person), *n* | 36, 13, 0, 2 | 30, 6, 3, 1 | 0.16 | — |

**Notes:**
Average (standard deviation).
SMI, skeletal muscle mass index.
* *p* value < 0.05.
Effect size indicated Cohen's d.

previous studies, and the measurement site was located 15 cm from the upper edge of the patella, considered to be strongly related to lower limb muscle strength (*Nakamura et al., 2019*), and was measured once with bare skin without covering. The thigh circumference thickness has sound reliability, and the ICC of measurements were 0.99–1.00 (*Kobayashi et al., 2018*).

## Statistical analysis

First, the normality of all variables was confirmed using the Shapiro–Wilk test. The attributes of the subjects were compared using the unpaired t-test and the chi-square test. Effect sizes were calculated using Cohen's d, for items for which significant differences were confirmed by comparing the two groups. Effect sizes greater than 0.80 are considered large and greater than 0.50 are considered moderate (*Cohen, 1988*). To confirm whether there is an effect of exercise frequency on the TMG parameters, the GLMM and Bonferroni's multiple comparison test were used with the TMG parameters as the dependent variable, sex as the without-subject factor, dominant/non-dominant foot as the within-subject factor, and exercise frequency as a covariate. Each parameter was compared using the GLMM and Bonferroni's multiple comparison test, factors being the same as before. Finally, the correlation between the TMG parameters and the other parameters was examined using Pearson's correlation coefficient for both males and females. All analyses were performed using the SPSS statistical package for Windows, version 27.0. software (IBM Corp., Armonk, NY, USA). Statistical significance was set at $p < 0.05$.

## RESULTS

The characteristics of the participants, including age, height, body mass, and skeletal muscle mass index (SMI) are shown in Table 1. There were significant differences in height, weight and SMI between male and female ($p < 0.001$), with effect sizes ranging from 0.79 to 11.69 and above moderate.

For the results of the GLMM on the Dm of the TMG parametors, the interaction (sex × dominant/non-dominant foot × exercise frequency) was not significant ($F = 3.35$, $p = 0.07$). The effect of exercise frequency as a covariate on Dm was not significant

**Table 2  ANOVA table for each parameter.**

| | | F value | Degrees of freedom | p value |
|---|---|---|---|---|
| Dm | sex | 3.29 | 1 | 0.07 |
| | dominant/non-dominant foot | 0.93 | 1 | 0.34 |
| | sex × dominant/non-dominant foot | 4.02 | 1 | 0.04* |
| Td | sex | 0.06 | 1 | 0.81 |
| | dominant/non-dominant foot | 0.24 | 1 | 0.63 |
| | sex × dominant/non-dominant foot | 0.65 | 1 | 0.42 |
| Tc | sex | 0.13 | 1 | 0.72 |
| | dominant/non-dominant foot | 0.54 | 1 | 0.47 |
| | sex × dominant/non-dominant foot | $F < 0.001$ | 1 | 0.99 |
| Lower limb muscle mass | sex | 152.62 | 1 | $p < 0.001$* |
| | dominant/non-dominant foot | 18.71 | 1 | $p < 0.001$* |
| | sex × dominant/non-dominant foot | 8.22 | 1 | 0.01* |
| Muscle thickness | sex | 64.95 | 1 | $p < 0.001$* |
| | dominant/non-dominant foot | 4.28 | 1 | 0.04* |
| | sex × dominant/non-dominant foot | 0.19 | 1 | 0.67 |
| Knee joint extension torque | sex | 17.28 | 1 | $p < 0.001$* |
| | dominant/non-dominant foot | 10.50 | 1 | 0.002* |
| | sex × dominant/non-dominant foot | 2.37 | 1 | 0.13 |
| Thigh circumference | sex | 18.02 | 1 | $p < 0.001$* |
| | dominant/non-dominant foot | 1.00 | 1 | 0.32 |
| | sex × dominant/non-dominant foot | 1.06 | 1 | 0.31 |

**Notes:**
Dm, displacement; Td, delay time; Tc, contraction time.
* $p < 0.05$.

($F = 1.69$, $p = 0.20$). For the results of the GLMM on Td, the interaction (sex × dominant/non-dominant foot × exercise frequency) was not significant ($F = 1.12$, $p = 0.29$). The effect of exercise frequency as a covariate on Td was not statistically significant ($F = 1.33$, $p = 0.25$). Regarding the results of GLMM on Tc, the interaction (sex × dominant/non-dominant foot × exercise frequency) was not significant ($F = 0.01$, $p = 0.94$). The effect of exercise frequency as a covariate on Tc was not statistically significant ($F = 0.55$, $p = 0.46$).

The GLMM results for sex, dominant foot, and non-dominant foot are presented in Table 2. The parameters used are listed in Table 3. In terms of the TMG parameters, for the results of the GLMM on the Dm, a main effect was not observed by sex ($F = 3.29$, $p = 0.07$) and dominant/non-dominant foot ($F = 0.93$, $p = 0.34$). An interaction was observed for Dm ($F = 4.02$, $p = 0.04$), and in the results of the multiple comparison test, no difference in Dm for the dominant foot, Dm for the non-dominant foot was significantly lower in females than in males. For the results of the GLMM on Td, a main effect was not observed by sex ($F = 0.06$, $p = 0.81$) and dominant/non-dominant foot ($F = 0.24$, $p = 0.63$). The interaction (sex × dominant/non-dominant foot) was not significant ($F = 0.65$, $p = 0.42$). For the results of the GLMM on Tc, a main effect was not observed by sex ($F = 0.13$, $p = 0.72$) and dominant/non-dominant foot ($F = 0.54$, $p = 0.47$). The interaction

**Table 3 Comparison of male and female parameters for dominant and non-dominant feet.**

| | Male (n = 51) | | | Female (n = 40) | | |
|---|---|---|---|---|---|---|
| | All | Dominant foot | Non-dominant foot | All | Dominant foot | Non-dominant foot |
| Dm, mm | 6.8 (1.9) | 6.6 (1.5) | 6.9 (2.2) | 6.1 (0.7) | 6.2 (1.9) | 6.1 (1.9)* |
| Td, ms | 26.4 (4.0) | — | — | 27.9 (5.4) | — | — |
| Tc, ms | 30.7 (5.6) | — | — | 28.7 (5.6) | — | — |
| Lower limb muscle mass, kg | 9.4 (1.4) | 9.4 (1.5) | 9.3 (1.4)† | 6.2 (0.9)* | 6.2 (0.9)* | 6.1 (0.9)*, † |
| Muscle thickness, mm | 20.9 (2.3) | 21.0 (2.3) | 20.7 (2.3) | 17.0 (2.6)* | 17.2 (2.6) | 16.7 (2.6) |
| Knee joint extension torque, Nm/kg | 2.53 (0.99) | 2.64 (1.09) | 2.42 (0.88) | 1.82 (0.55)* | 1.86 (0.59) | 1.79 (0.53) |
| Thigh circumference, cm | 49.8 (4.1) | — | — | 46.6 (2.8)* | — | — |

Notes:
Dm, displacement; Td, delay time; Tc, contraction time, average (standard deviation) [95% Confidence interval]; *, male *vs* female; †, dominant *vs* non-dominant foot.*,
†: $p < 0.05$.

(sex × dominant/non-dominant foot) was not significant ($F < 0.001$, $p = 0.99$). In terms of the lower limb muscle mass, a main effect (sex: $F = 152.62$, $p < 0.001$, dominant/non-dominant foot: $F = 18.71$, $p < 0.001$) and an interaction (sex × dominant/non-dominant foot) were observed ($F = 8.22$, $p = 0.01$), and in the results of the multiple comparison test, values of the females were significantly lower than males. In both sexes, the values in the non-dominant foot were significantly lower than in the dominant foot. In terms of muscle thickness, a main effect was observed (sex: $F = 64.95$, $p < 0.001$, dominant/non-dominant foot: $F = 4.28$, $p = 0.04$), an interaction (sex × dominant/non-dominant foot) was not significant ($F = 0.19$, $p = 0.67$), and the females were significantly lower than males. In terms of knee joint extension torque, a main effect was observed (sex: $F = 17.28$, $p < 0.001$, dominant/non-dominant foot: $F = 10.50$, $p = 0.002$), an interaction (sex × dominant/non-dominant foot) was not significant ($F = 2.37$, $p = 0.13$), and the females were significantly lower than males. In terms of thigh circumference, a main effect was observed by sex ($F = 18.02$, $p < 0.001$), was not observed by dominant/non-dominant foot ($F = 1.00$, $p = 0.32$). The interaction (sex × dominant/non-dominant foot) was not significant ($F = 1.06$, $p = 0.31$), and values for females were significantly lower than males.

Table 4 presents the correlation coefficients between the TMG parameters and lower-limb muscle mass, muscle thickness, joint torque, and thigh circumference for males and females. Dm was negatively correlated with lower-limb muscle mass in both sexes. Meanwhile, Td was positively correlated with lower-limb muscle mass only in males and negatively correlated with muscle thickness only in females. Tc was positively correlated with lower-limb muscle mass only in females.

## DISCUSSION

In the TMG parameters, only Dm showed an interaction. The results of the GLMM highlighted that sex and dominant/non-dominant foot are associated with Dm for TMG parameters in healthy college students. These results support some of our first research hypotheses on Dm and support our research hypothesis on Td and Tc. Based on the multiple comparison test, upon examination of the dominant and non-dominant feet, Dm

**Table 4 Correlation coefficient between TMG parameters and each parameter.**

|  |  | Lower limb muscle mass | Muscle thickness | Knee joint extension torque | Thigh circumference |
|---|---|---|---|---|---|
| Dm | Male | −0.24* | 0.01 | 0.12 | 0.16 |
|  | Female | −0.29* | −0.09 | 0.06 | −0.22 |
| Td | Male | 0.38* | −0.01 | −0.07 | 0.14 |
|  | Female | 0.19 | −0.21* | −0.15 | 0.07 |
| Tc | Male | −0.004 | −0.03 | −0.11 | 0.03 |
|  | Female | 0.20* | 0.06 | 0.02 | 0.12 |

Notes:
Dm, displacement; Td, delay time; Tc, contraction time.
* $p < 0.05$.
For the other parameters, the values for each dominant and non-dominant feet were aligned and used in the analysis for each gender for the TMG parameters.

was the only TMG parameter that was different between males and females for the non-dominant foot, indicating that Dm is the most likely indicator of sex difference in TMG. This result highlights the possibility that in healthy college students, females have a smaller amount of vertical displacement during muscle contraction due to electrical stimulation. Several studies have reported sex differences in the TMG parameters (*Šimunič et al., 2018*; *Simunic et al., 2017*). No sex differences were found in Td and Tc, particularly in the rectus femoris muscles of college students who exercise infrequently, such as the subjects in this study. Td and Tc assess the physiological components of neuromuscular function. The vastus lateralis muscles of 9- to 14-year-old children have higher Tc in boys than in girls (*Simunic et al., 2017*), which is consistent with reports of a higher proportion of type I fibers in the vastus lateralis in 16-year-old boys than in girls (*Glenmark, Hedberg & Jansson, 1992*). The rectus femoris muscle is known to have a high percentage of type II fibers (approximately 60–70%) and low percentage of type I fibers (approximately 30–40 %) (*Johnson et al., 1973*). Changes in muscle fiber type are caused by aging, exercise, and immobility (*Carmeli & Reznick, 1994*). The subjects of this study, college students, were young, approximately 20 years old, and although they exercised infrequently, they were independent in their daily lives. Therefore, there was no difference in the Td and Tc of the rectus femoris muscle, which has many type II fibers, between male and female university students. These TMG parameter results differ from the evaluation of muscle function frequently used in clinical practice, such as muscle mass and strength. Therefore, when using the Dm of the TMG as an evaluation battery in the rectus femoris muscle, it is necessary to consider sex-based differences when choosing subjects.

Overall, there was a weak correlation (correlation coefficient: 0.20–0.38) between the parameters, with a significant correlation between the TMG parameters and each parameter. Specifically, there was a correlation between multiple TMG parameters and lower-limb muscle mass. The Dm and lower limb muscle mass in men (correlation: −0.24) and women (correlation: −0.29) were negatively correlated. These results support our second research hypothesis that lower-extremity muscle mass is negatively correlated with Dm. In addition, there was a correlation of 0.38 for the Td and lower limb muscle mass in men, and a correlation of 0.20 for the Tc and lower limb muscle mass in women. These results partially support our second research hypothesis. These results indicate that

lower-extremity muscle mass may be related to the neuromuscular responses of muscles in healthy college students. In TMG parameters, a decrease in Dm corresponds to an increase in muscle stiffness (*García-García et al., 2019*; *Šimunič et al., 2019*). Td reflects the responsiveness of muscles to contraction due to electrical stimulation (*García-García et al., 2019*). Tc reflects the contraction rate from the onset to the end of muscle contraction. Therefore, the negative correlation between lower extremity muscle mass and Dm may indicate that those with high muscle mass in the lower limbs have lower Dm, that is, higher muscle stiffness. It is possible that in healthy college students with low muscle mass of the lower limbs, muscle responsiveness to contraction is quicker, whereas those with a large amount of lower limb muscle may require longer times for muscle contraction from electrical stimulation.

There was a correlation of −0.20 for the Td and muscle thickness in women. TMG parameters and muscle thickness represent the function of the rectus femoris muscle, lower limb muscle mass represents the muscle mass of the entire lower limb, and joint torque and thigh circumference represent the muscle function of the extensor group of the knee joint. This may indicate that the greater thickness of the rectus femoris muscle is more responsive to muscle contraction. A previous study reported that bed rest induced changes in both Dm and Tc of the TMG signal in several lower extremity muscles, and changes in Dm were inversely related to those of muscle thickness, especially in the gastrocnemius muscle (*Pišot et al., 2008*). Since this present study was a cross-sectional study and the relationships were investigated in the rectus femoris muscle alone, it may not fully reflect the characteristics of each parameter. Since muscle thickness did not significantly correlate with Dm or Tc in the present study and the rectus femoris muscle was not studied in the previous study, future studies on multiple muscles, and longitudinal studies, are needed. In the subjects in this study, the effect of exercise frequency on TMG parameters was examined in the GLMM; however, previous exercise history or type of exercise was not known. In general, people with more exercise experience tend to have a greater muscle mass and better neuromuscular function. In college students who exercise infrequently, muscle mass, exercise history, or type of exercise may also need to be considered when conducting longitudinal studies on TMG.

In a report on the validity of the TMG parameter, maximum muscle strength and muscle fatigue rate were often verified. A high correlation has been reported (*Raeder et al., 2016*; *Hunter et al., 2012*), but in this study, the maximum muscle strength measured by the manual muscular strength meter and femoral circumference had little to do with TMG parameters. This may be due to differences in muscle strength measurement methods or differences in participants who do not play sports on a daily basis. Because the evaluation of muscle functions, such as lower limb muscle mass and joint torque, does not include individual muscles but involves multiple muscle functions working in combination, this is thought to reflect the amount of activity of daily life (*Hasegawa et al., 2008*). In sports and medicine, muscle strength, thickness, fatigue rate, and risk of injury, including muscle physiology and pathophysiology, are substantially affected by sex (*Fujisawa et al., 2017*; *Hill et al., 2018*). Similarly, in this study, lower limb muscle mass, muscle thickness, knee joint extension torque and thigh circumference differed between men and women.

Although many participants in our study did not exercise on a daily basis, they were generally healthy college students. Muscle mass may be the most relevant TMG parameters in healthy college students. At present, information on the application of TMG parameters in healthy college students and females is limited, making it difficult to determine their characteristics by comparing the findings of previous studies on TMG in athletes (García-García et al., 2019; Lohr et al., 2019). The results of this study provide basic information for future application of TMG to patients who would attend a hospital. Therefore, TMG may be a unique evaluation method for assessing neuromuscular function.

## LIMITATIONS

This study has some limitations. Because this was a cross-sectional study, the relationship between each item in the TMG parameter, type of exercise, and motor function was unclear. As only the rectus femoris muscle was examined in this study, it is possible that the relationship with the medial, intermediate, and lateral vastus muscles, which are monoarticular knee extensors, may not be similar. It will be necessary to comprehensively investigate sex, type of exercise, and motor function in the future.

## CONCLUSIONS

In healthy college students, TMG parameters for the rectus femoris showed sex differences in Dm, and there was a weak correlation between TMG parameters and lower limb muscle mass, but not with muscle function evaluation conventionally used in clinical practice. Practitioners in the field of sports medicine must understand that TMG parameter evaluation differs from muscle function evaluation, which is conventionally used in clinical practice. When using the Dm of the TMG as an evaluation battery for the rectus femoris muscle, it is important to consider sex-related differences. The TMG parameter is a unique evaluation method for neuromuscular function assessment.

## ACKNOWLEDGEMENTS

We would like to thank Editage for the English language editing. The authors would like to thank all the students who participated in this study.

### Funding

The authors received no funding for this work.

### Competing Interests

Hayato Goto is employed by LITALICO lnc. The authors declare that they have no competing interests.

## Author Contributions

- Yasuaki Kusumoto conceived and designed the experiments, analyzed the data, prepared figures and/or tables, authored or reviewed drafts of the article, and approved the final draft.
- Hayato Goto conceived and designed the experiments, performed the experiments, authored or reviewed drafts of the article, and approved the final draft.
- Kohei Chiba conceived and designed the experiments, performed the experiments, authored or reviewed drafts of the article, and approved the final draft.
- Sakiko Oonishi conceived and designed the experiments, analyzed the data, prepared figures and/or tables, authored or reviewed drafts of the article, and approved the final draft.
- Junko Tsuchiya conceived and designed the experiments, analyzed the data, prepared figures and/or tables, authored or reviewed drafts of the article, and approved the final draft.

## Human Ethics

The following information was supplied relating to ethical approvals (*i.e.*, approving body and any reference numbers):

The Tokyo University of Technology granted ethical approval to conduct the study within its facilities (authorization number: E16HS-030).

## Data Availability

Data are available at Figshare:

Yasuaki, Kusumoto (2021): Characteristics of muscle contraction of the rectus femoris using Tensiomyography by sex in healthy adolescents. figshare. Dataset. https://doi.org/10.6084/m9.figshare.16945522.v1.

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
