# Peer review of "Characteristics of muscle contraction of the rectus femoris using tensiomyography by sex in healthy college students: a cross-sectional study"

_PeerJ, doi:10.7717/peerj.14732_

## Round 0.1 · original submission · Major Revisions

Dear authors,

Please follow the reviewers' comments carefully and respond to the reviewers point by point.

Reviewer 1 ·

Basic reporting

The article is written in fairly clear English, but there is some vagueness and ambiguity throughout, particularly: greater details should be provided to explain the methodologies used (examples: the use of "adolescent" ("adolescence") is inappropriate, the World Health Organisation classification of adolescence is 10-19 years old; L66-68 the rationale for the study is based around geriatric medicine, yet the study was conducted in young healthy individuals; L76-78 it is not clear why TMG could be considered as a potential substitute for other assessments used in clinical practice; L79-81 the justification for assessing rectus femoris is not apparent; L114-138 more details about the TMG protocol are required [stimulator and electrode models; software version number; knee joint angle and whether the limb was secured in place or free; pulse shape; inter-stimuli interval); L140-145 more details are needed on how body composition is determined, especially regarding different body segments; L174-180 effect sizes are reported in the results section, these should be explained in the materials and methods.

The hypothesis is vague, the specific "differences" in TMG parameters that were anticipated should be reported and justified by a stronger rationale created throughout the introduction section.

The authors suggest that there is no prior research using TMG in clinical settings or amongst young people (L60-68) but have overlooked a number of research articles, notably: Šimunič, B., Koren, K., Rittweger, J., Lazzer, S., Reggiani, C., Rejc, E., ... & Degens, H. (2019). Tensiomyography detects early hallmarks of bed-rest-induced atrophy before changes in muscle architecture. Journal of applied physiology, 126(4), 815-822.; Pišot, R., Narici, M. V., Šimunič, B., De Boer, M., Seynnes, O., Jurdana, M., ... & Mekjavić, I. B. (2008). Whole muscle contractile parameters and thickness loss during 35-day bed rest. European journal of applied physiology, 104(2), 409-414.; and Simunic, B., Degens, H., Zavrsnik, J., Koren, K., Volmut, T., & Pisot, R. (2017). Tensiomyographic assessment of muscle contractile properties in 9-to 14-year old children. International Journal of Sports Medicine, 38(09), 659-665.

Experimental design

The research falls within the aims and scope of PeerJ, however there are some major issues which call into question the validity of the data: given that strength is the only (potentially) fatiguing measurement performed - isometric strength test is in fact associated with minimal fatigue - the randomization of measurements is problematic here, a standardised rest time before each measure would have been more important to permit valid measurements to be performed - it would seem that this standardization has been compromised by the randomization of measurement order (L109-111).

As described under Basic Reporting, the methods need greater description; further examples include: L94-98 how was the sample size determined to ensure appropriate statistical power for the study?; ICC is reported for TMG variables, similar reporting for other measurements (in particular ultrasound) would have been welcome; L166 body mass was presumably measured as part of the study, this should be reported.

Validity of the findings

The rationale for the study and the benefit to the literature should be more-firmly established (see Basic Reporting).

Some weak correlations are reported between parameters determined using TMG and anthropometric/ functional assessments, but the discussion section does not explain these relationships. The selection of assessments (measuring only a single muscle head [rectus femoris] using TMG, to relate to knee extension strength and/ or thigh circumference, for example) is questionable (L234-237).

Additional comments

No further comments

·

Basic reporting

Authors report sex- and site- effects on TMG-derived parameters of RF muscle, quadriceps girth, lower limb muscle mass, knee extension MVC, RF muscle thickness. In cross-sectional study of 91 university students, both males and females. Thex confirmed sex- affect in all outcomes; where as in TMG Dm also an interaction effect. Furthermore, they reported Pearson r between TMG-derived parameters and other outcomes.

Main considerations:
- Statistics: I see dominant/non-dominant site as a within factor. There is clear dependency among both sites. This might change all results and discussion. Therefore, only general comments will be put forward.
- It is unclear which effect size was reported. You should explain this in Methods->statistics and omit reporting effect size if N.S.
- When there is no site main effect or sex*site effect you should not do post-hoc analysis in each site separately. You should merge both sites, e.g. average.
- TMG does not measure in vertical directions (abstract) but transversal to muscle fibers.
- many publication compared TMG in healthy males and females (see DOI: 10.1093/gerona/gly069). You should appreciate those.
- Garcia-Garcia (2019) could not be (alone) a reference for Dm-Stiffness. At least two original articles linked Dm with atrophy induced stiffness loss (DOI: 10.1152/japplphysiol.00880.2018; DOI: 10.1152/japplphysiol.00880.2018)
- Again, Garcia-Garcia (2019) should not be a citation for Tc-Muscle composition. Use those articles, instead (DOI: 10.1249/MSS.0b013e31821522d0; DOI: 10.1016/j.jbiomech.2004.10.020)
- Report more details on RF thickness assessment. Distances?
- What was the knee angle in MVC assessment. Was effort explosive or a ramp?
- Was circumference measure in standing position (support, not supported?) or lying position?
- use p rather than P for significance
- Results should be rewritten reporting all findings. Not just stating where the results are.
- i would suggest to pool dom/nondom data as average for all measures where there were no main site effects nor site*sex interaction effect.

Experimental design

- You've specified that randomisation was used among measure. But MVC requires warm-up. How was warm-up integrated in this design as it is well known warm-up affects some of the outcomes.
- Electrodes were most likely not put around the sensor, but rather proximal and distal.

Validity of the findings

- clarification for warm-up effect is needed.
- Site as within factor is recommended.
- Averages should be used as recommended above

Additional comments

I would wait with reviewing discussion as the results might change a lot when site will be considered as within factor and site averages used for post-hoc analysis.

Reviewer 3 ·

Basic reporting

Dear authors, although I like the purpose of your manuscript and I think your results could be of common interest. However, too many improvements are needed because the manuscript is incoherent in several parts.
As your title reports, you do expect to find information on tensiomyography results in adolescents, while the study was conducted in younger adults (university students). In lines 63-68 you explained the importance of muscle variation in geriatric medicine and adult, but your study analyzed youths. No previous studies on younger were shown. It is not clear what kind of information I should be expected to find. You could improve this section and make your introduction more coherent with your research. So, appropriate references should be cited. Also, what are previous results of muscle sex differences assessed by TMG? In addition, what did you expect to find? It seems obvious to find differences between sexes due to anatomical and anthropometrical, physiological, hormonal, and others characteristics gender-related.

Experimental design

The aims of your research are very interesting but in my opinion, your methods lack many details.
1) A priori sample size calculation or post-doc statistical power calculations should be done.
2) In lines 109-111 you reported "Considering the effect of fatigue...Excel". A randomized measurement order does not allow to exclude a fatigue influence. Each participant should be previously instructed to perform an unbiased evaluation (for example no high-intensity exercise for 48h before the test, etc.)
3) Body composition may be improved. What BC parameters did you analyze? Which estimation functions did you use? Only height and weight are reported in tables. In my opinion, you should specify these details.
4) The maximum thigh circumference estimation should be better described. You can consult "Anthropometric Standards for the Assessment of Growth and Nutritional Status" by R. Frisancho for specific details.
5) In the statistical analysis paragraph you should report descriptive statistics as mean and standard deviation (i.e. as reported in Table 1).

Validity of the findings

In my opinion, too many confounding factors should influence your results as physical activity or sports practice, anthropometric and body composition differences, and nutrition. Your results should be corrected by these characteristics.
Also, significant differences are presented only for sex analysis, but it is expected due to gender characteristics. Since the study design is cross-sectional, you should better interpret and justify your results in the discussion paragraph using appropriate physiological and methodological references.

---

## Round 0.2 · Minor Revisions

Please respond to reviewer comments.

Reviewer 1 ·

Basic reporting

The authors have extensively reworked the manuscript in accordance with reviewer recommendations. However, some points have been misinterpreted or only partially addressed.

L258-260: As requested, a line about effect sizes has been added to the methods section, however this sentence does not describe what exact effect size calculation was used (this is reported as Cohen's d in tables) and how effect sizes were interpreted.

L181-182: It is not clear what is meant by radial displacement being measured "vertically", this line should be amended to reflect what is reported within the abstract ("maximum displacement of the muscle belly in the radial direction with respect to the muscle"), which is more accurate.

L270-299: Statistical outputs have only been reported for some analyses, p-values have been included in tables, but outputs should be fully reported within the text also.

Experimental design

The revised details expand on the experimental design, which is now more clearly described.

Validity of the findings

Re-analysis of data has been performed in accordance with reviewers' suggestions, however full reporting of statistical outputs (see Basic reporting) is necessary to allow the reader to assess the validity of the findings.

Additional comments

No further comment.

---

## Round 0.3 · accepted · Accept

Dear authors,

Thank you for your replies!

In the present form, the paper is ready for publication.
Congratulations.